# Porous Ceramics Adsorbents Based on Glass Fiber-Reinforced Plastics for NO_x_ and SO_x_ Removal

**DOI:** 10.3390/polym14010164

**Published:** 2021-12-31

**Authors:** Hiroyuki Kinoshita, Kentaro Yasui, Taichi Hamasuna, Toshifumi Yuji, Naoaki Misawa, Tomohiro Haraguchi, Koya Sasaki, Narong Mungkung

**Affiliations:** 1Department of Engineering, University of Miyazaki, 1-1 Gakuen-Kibanadai-Nishi, Miyazaki 889-2192, Japan; haraguti@cc.miyazaki-u.ac.jp; 2National Institute of Technology, Kagoshima College, 1460-1 Hayatochoshinko, Kirishima City 899-5193, Kagoshima Prefecture, Japan; yasui@kagoshima-ct.ac.jp; 3Graduate School of Engineering, University of Miyazaki, 1-1 Gakuen-Kibanadai-Nishi, Miyazaki 889-2192, Japan; hj16044@student.miyazaki-u.ac.jp; 4Faculty of Education, University of Miyazaki, 1-1 Gakuen-Kibanadai-Nishi, Miyazaki 889-2192, Japan; yuji@cc.miyazaki-u.ac.jp; 5Center for Animal Disease Control, University of Miyazaki, 1-1 Gakuen-Kibanadai-Nishi, Miyazaki 889-2192, Japan; a0d901u@cc.miyazaki-u.ac.jp; 6Suzuki Motor Corporation, 300 Takatsuka-cho, Minami-ku, Hamamatsu City 432-8611, Shizuoka Prefecture, Japan; road15024@outlook.jp; 7Department of Electrical Technology Education, King Mongkut’s University of Technology Thonburi, Bangkok 10140, Thailand; narong_kmutt@hotmail.com

**Keywords:** waste GFRP, reuse, adsorbent, ceramics, NO_2_ adsorption performance, SO_2_ adsorption performance

## Abstract

To reuse waste glass fiber-reinforced plastics (GFRPs), porous ceramics (i.e., GFRP/clay ceramics) were produced by mixing crushed GFRP with clay followed by firing the resulting mixture under different conditions. The possibility of using ceramics fired under a reducing atmosphere as adsorbent materials to remove NO_x_ and SO_x_ from combustion gases of fossil fuels was investigated because of the high porosity, specific surface area, and contents of glass fibers and plastic carbides of the ceramics. NO_2_ and SO_2_ adsorption tests were conducted on several types of GFRP/clay ceramic samples, and the gas concentration reduction rates were compared to those of a clay ceramic and a volcanic pumice with high NO_2_ adsorption. In addition, to clarify the primary factor affecting gas adsorption, adsorption tests were conducted on the glass fibers in the GFRP and GFRP carbides. The reductively fired GFRP/clay ceramics exhibited high adsorption performance for both NO_2_ and SO_2_. The primary factor affecting the NO_2_ adsorption of the ceramics was the plastic carbide content in the clay structure, while that affecting the SO_2_ adsorption of the ceramics was the glass fiber content.

## 1. Introduction

Currently, most glass fiber-reinforced plastic (GFRP) wastes are sent to landfills without recycling because they are difficult to recycle using the existing recycling technologies due to the content of glass fibers [1,2]. Leachates from landfills are causing several health and environmental problems. Thus, an efficient waste GFRP recycling technology is required [3,4,5,6].

In previous studies, authors have proposed a method for waste GFRPs reuse involving the production of porous ceramics by mixing crushed waste GFRPs with clay and firing the resulting mixture [7,8,9]. The types of clay and GFRPs used for the ceramic production are selected according to the type of the required products. In the ceramic manufacturing process, the resin components in GFRPs thermally decompose during the firing process, and the glass fibers remain in the clay matrix. The generation of fine glass fiber dust is reduced during this process because both the clay and glass fibers or the glass fibers alone are sintered during the firing process of ceramics [8,9].

Focusing on the high porosity of GFRP/clay ceramics, we aimed to develop ecofriendly products [10,11,12,13,14]. Ceramics produced by firing the mixture of GFRPs and clay powders under an oxidizing atmosphere exhibit high porosity and a relatively high specific strength because the clay matrix is reinforced by the glass fibers. Moreover, this type of ceramics can also have high permeability by adjusting the particle size and the mixing ratio of GFRPs in the manufacturing process. These ceramics were investigated as filtering materials for turbid water treatment [12] and water-permeable paving blocks [13].

However, ceramics produced by firing the mixture of GFRPs and clay powders under a reducing atmosphere contain a small amount of plastic carbide residue with the glass fibers. Consequently, these ceramics exhibit higher specific surface area than those produced under an oxidizing atmosphere [14,15]. Because of the high specific surface area and porosity of the ceramics as well as the high ion exchange function of the clay [16,17,18,19], this type of ceramics was studied as a dye adsorbent material to remove pigments from dye-contaminated wastewater [14,15].

The GFRP/clay ceramics produced under reducing conditions may also be used as an adsorbent for toxic gases because the performance of a gas adsorbent is typically linked to its high specific surface area and porosity [20]. Further, they contain glass fibers and plastic carbides with fine pores. Carbides with fine pores or activated carbons exhibit high adsorption for specific substances [21,22,23,24,25,26,27]. Glass fibers also contain a large amount of calcium, which easily reacts with sulfur oxides [28]. Thus, the authors investigated the possibility of using GFRP/clay ceramics as adsorbent materials to remove NO_x_ and SO_x_ from gas emissions from fossil fuel combustion. In the study, the authors aimed to develop ceramics with a high adsorption ability for both NO_x_ and SO_x_ gas [29].

During the waste incineration process, several techniques, such as combustion control as well as non-catalytic and catalytic denitrification equipment, are used to reduce or remove NO_X_ from combustion gases. Combustion control reduces the generation of nitrogen oxides by adjusting the combustion conditions of the wastes in the incinerator to suppress the generation of incomplete combustion gases. The non-catalytic denitrification equipment reduces nitrogen oxides by directly spraying ammonia gas into the incinerator. Catalytic denitrification reduces NO_x_ to N_2_ via a catalytic reaction [30,31,32].

SO_x_ are removed from the combustion gases of fossil fuel by either the dry method or the wet method. In the wet method, sulfur oxides are collected by spraying a large amount of an alkaline solution, such as caustic soda, into the incinerator to form a solution, such as Na_2_SO_4_. The dry method involves (1) spraying an alkaline agent, such as slaked lime, into the incinerator to neutralize the acidic substances in the exhaust gas and (2) collecting the reaction products as fly ash [33]. Moreover, NO_x_ can be reduced using zeolites [34,35], activated carbons, and photocatalytic technology [36]. Several studies, such as that of Kitagawa et al., investigated the production of activated carbon from plastics [37].

Here, the effect of the specific surface area as well as the plastic carbides and glass fiber contents on the NO_x_ and SO_x_ adsorption performance of GFRP/clay ceramics were studied using several types of ceramic samples produced by adjusting the GFRP mixing ratio and the firing atmosphere. Next, NO_2_ and SO_2_ adsorption experiments were conducted using the prepared ceramic samples, and the reduction rates of their concentration were compared to those treated using a clay ceramic and Bora, a volcanic pumice stone, with a high NO_2_ adsorption ability [38]. Furthermore, to determine the primary factor affecting NO_x_ and SO_x_ adsorption, adsorption tests were conducted on the glass fibers in GFRPs and GFRP carbides. NO_2_ and SO_2_ adsorption performances of the GFRP/clay ceramics were evaluated based on the experimental results. In addition, their adsorption mechanisms were investigated based on the difference in the contents of plastic carbide and glass fibers in the ceramics and their specific surface area.

## 2. Materials and Methods

### 2.1. Samples Used for NO_2_ and SO_2_ Adsorption Tests

Several types of oxidatively and reductively fired GFRP/clay ceramics, a clay ceramic (without glass fibers), and unfired Bora were used for the NO_2_ and SO_2_ adsorption tests.

Table 1 presents the inorganic chemical compositions of GFRP, clay and Bora [14,15]. Their compositions were measured by an energy dispersive X-ray analyzer (EDX-720, Shimadzu Corporation, Kyoto, Japan) using a fundamental parameter method after firing at 1073 K.

Polyamide (PA) thermoplastic pellets (Renny, Mitsubishi Engineering-Plastics Co., Tokyo, Japan) containing glass fiber (~40 mass%) were used as a substitute for waste GFRPs. The glass fibers in the GFRPs were E-glass type fibers with diameters of ~10 µm, lengths of ~1.0 mm, and a high CaO content.

The clay (Sougoo Co., Miyakonojo, Japan) was mined and produced in Miyazaki, Japan [39]. It is typically used as a raw material for bricks and tiles. The clay comprises chlorite group as major minerals.

Similar to the clay, the volcanic pumice Bora (Nanken kougyo Co., Miyakonojo, Japan) was mined in Miyazaki, Japan. Bora is mined in large quantities in the southern Kyushu area of Japan. The chemical composition of Bora was very similar to that of the clay although it had a slightly higher calcium content.

Figure 1 shows the manufacturing process of the GFRP/clay ceramic samples. The sample manufacturing procedure was as follows [14,15]:The clay was crushed using a rotary mill (New Power Mill ABS-W, Osaka Chemical Co., Ltd., Osaka, Japan) and was then sifted using a 0.3 mm mesh screen.GFRPs were also crushed using the same rotary mill and were then sifted using a 0.5 mm mesh screen.The crushed GFRPs were mixed with the clay at the mass rates of 20%, 40%, and 60%.The GFRP/clay mixture was solidified by being pressed into a mold under 10 MPa. The molded samples had a diameter of 74 mm and a thickness of 50–60 mm.The molded samples were heated under oxidizing or reducing atmospheres to 1073 K in an electric furnace (KY-4N, Kyoei Electric Kilns Co., Ltd., Tajimi, Japan). The samples were then held at the firing temperature for 1 h and then allowed to cool to room temperature in the furnace. For oxidative firing, the samples were heated at 100 K h^−1^ to the firing temperature. For reductive firing, the samples were heated at 400 K h^−1^. The reducing atmosphere was obtained by closing the intake port attached to the bottom of the electric furnace.The produced GFRP/clay samples were then crushed using a hammer, and the particles with sizes of 1.4–2.0 mm were selected.

Glass fibers and GFRP carbides were prepared to be used in the experiments investigating the factors affecting the gas adsorption for GFRP/clay ceramics. The glass fibers were prepared by heating the GFRP pellets at a rate of 100 K h^−1^ to 1073 K under an oxidizing atmosphere followed by holding them at the firing temperature for 1 h until the plastic component decomposed. The GFRP carbides were prepared by heating the GFRP pellets at a rate of 400 K h^−1^ to 1073 K under a reducing atmosphere. The particle size of both samples was adjusted to 1.4–2.0 mm by sieving.

Figure 2 shows examples of microscope (SZX10, Olympus Corporation, Tokyo, Japan) images of the samples [14,15]. The reductively fired GFRP/clay ceramic samples were black owing to the plastic carbide residues.

Table 2 presents the inorganic chemical compositions of the samples [14,15]. The GFRP/clay ceramics had greater CaO contents than the clay ceramic, and the ratio of CaO to the total mass of the ceramics increased with the GFRP mixing ratio because of the increase in the glass fiber content.

### 2.2. Material Properties of the Samples

The following material properties of GFRP/clay ceramics have already been reported [14,15]. However, they are shown here since it is necessary to discuss the NO_x_ and SO_x_ adsorption performance of the ceramics in terms of their properties.

Table 3 [14,15] presents the apparent porosity, specific surface area, and carbon content of the samples. The apparent porosity was measured using a mercury porosimeter (Auto Pore IV 9500, Micromeritics Instrument Corporation, Norcross, GA, USA). The specific surface area was measured using a high-precision gas and vapor adsorption analyzer (BELSORP-max, MicrotracBEL Corp., Osaka, Japan). The carbon content was measured using an elemental analyzer (CHNS/O Analyzer 2400, PerkinElmer Inc., Waltham, MA, USA).

The apparent porosities of the oxidatively and reductively fired GFRP/clay ceramics were equivalent and were approximately twice that of the clay ceramic because the resin component decomposes during the firing to create voids in the clay structure.

The specific surface areas of the oxidatively fired GFRP/clay ceramics were smaller than that of the clay ceramic and decreased with the increase in the mixing ratio of GFRPs. By contrast, the specific surface areas of the reductively fired GFRP/clay ceramics were comparable to that of the clay ceramic or slightly larger. The specific surface area of Bora was slightly lower than that of the clay ceramic. GFRP carbides exhibited a remarkably high specific surface area.

The carbon content in the reductively fired GFRP/clay ceramics were higher than that in the oxidatively fired ceramics because plastic carbides remained in the ceramic structure after the firing process. The carbon content in the GFRP pellets used as the raw material for the ceramics was ~30%, and the content in the GFRP carbides was 7%.

Figure 3 [14,15] shows the pore size distributions in the samples. The pore size distributions were measured using the same high-precision gas/vapor adsorption measurement instrument, which was used to measure the specific surface area. The nano-sized pores in the structure of the oxidatively fired GFRP/clay ceramics decreased with the increase in the mixing ratio of GFRPs. Thus, the specific surface areas of the ceramics decreased with the increase in the mixing ratio of GFRPs. The nano-sized pores could have disappeared owing to the progress of sintering between the clay and glass fibers with the increase in the mixing ratio of GFRPs.

The reductively fired GFRP/clay ceramics had relatively more nano-sized pores than the oxidatively fired GFRP/clay ceramics. Therefore, the ceramics had higher specific surface areas. This can be attributed to the high specific surface area of the GFRP carbides in the clay structure of the ceramics. In addition, the sintering between the clay and glass fibers may be suppressed owing to the presence of plastic carbides. The GFRP carbides had many pores with a size of several tens of nanometers.

Figure 4 shows examples of scanning electron microscope (SEM, S5500, Hitachi High-Technologies Corporation, Tokyo, Japan) images of the surface structures of the clay and 40% GFRP/clay ceramic samples. Further, the GFRP/clay ceramics had more pores than clay ceramic.

The NO_2_ and SO_2_ adsorption performance of the samples with the above material properties was examined.

### 2.3. Methodology of NO_2_ and SO_2_ Adsorption Tests

Figure 5 shows a schematic diagram of the NO_2_/SO_2_ adsorption test. The experimental apparatus was constructed using a standard NO_2_/SO_2_ gas cylinder; a standard air cylinder (Sumitomo Seika Chemicals Co., Ltd., Osaka, Japan); a 50 L gas storage bag (chemical resistance sampling bag (Tedlar bag), Tech-Jam, Osaka, Japan); an air pump (APN-085E-D2-W, IWAKI, Tokyo, Japan); a digital flow meter (CMS0020BSRN, Azbil, Tokyo, Japan); a test tube, into which the samples were placed; and an instrument for measuring the NO_2_ or SO_2_ concentrations (XPS-7, New Cosmos Electric, Osaka, Japan). Standard gases are usually used for calibrating the gas concentration. The standard gases are manufactured in accordance with the standards of Japan Calibration Service System (JCSS) [40,41,42,43]. The standard air gas has a density of 1.20 kg/m^3^ and a humidity of 65% (RH) at 293 K and 101.325 kPa. The NO_2_ and SO_2_ adsorption tests were performed following the subsequent procedures.

Samples were washed with distilled water and dried in an electric furnace at 378 K for over 24 h before the gas adsorption tests.In the NO_2_ adsorption tests, a 50 L gas storage bag was filled with NO_2_ gas (20 L, NO_2_ concentration = ~5 vol ppm).In the SO_2_ adsorption tests, a 50 L gas storage bag was filled with SO_2_ gas (10 L, SO_2_ concentration = ~10 vol ppm) and standard air (10 L). To homogenize the concentration of the gas mixture in the gas storage bag, the gas mixture was circulated in the circuit at a flow rate of 2 L/min for 20 min.A portion of the sample (5 g) was placed into a test tube. The NO_2_ or SO_2_ gas was allowed to pass through the test tube containing the sample at a flow rate of 1.0 L/min, and the gas was circulated in the circuit for a maximum of 4 h.The NO_2_ and SO_2_ concentrations in the gas storage bag were measured at 30 min intervals. The pump was momentarily stopped during the measurement of the gas concentration. The room temperature during the experiments was 287 K–299 K.

## 3. Results

### 3.1. NO_2_ Adsorption Performance of GFRP/Clay Ceramics

Figure 6 shows the NO_2_ reduction rates for all samples. Each graph represents the average reduction rates calculated based on the measurements of the three samples. Error bars represent the upper and lower limits of the measured values. The NO_2_ reduction rate of the Bora is shown in all figures for easy comparison with that of each sample. The graph marked “without sample” shows the NO_2_ reduction rate when the sample was not placed into a test tube. In other words, the graph represents the natural reduction in the NO_2_ concentration with time. The error in the NO_2_ reduction rate in the test without the sample is assumed to be mainly due to the difference in humidity caused by the difference in air temperature.

The NO_2_ reduction rates of the oxidatively fired GFRP/clay ceramics were roughly comparable to that of the Bora, although those of 40% and 60% GFRP/clay ceramics were slightly lower. The reduction rate of the clay ceramic was almost comparable to that of the Bora. The NO_2_ reduction rates of the reductively fired GFRP/clay ceramics were considerably higher than those of the oxidatively fired ceramics or the Bora regardless of the GFRP mixing ratio. The NO_2_ reduction rates of reductively fired GFRP/clay ceramics were higher than those of a porous concrete sample coated with titanium oxide, which has a high NO_x_ reduction rate reported in a previous study [39].

The above results demonstrate that the reductively fired GFRP/clay ceramics had a high adsorption capacity for NO_2_. This suggests that the high specific surface area of the ceramics had a significant effect on the NO_2_ adsorption performance or that plastic carbides in the ceramic structure are good NO_2_ adsorbents.

### 3.2. SO_2_ Adsorption Performance of GFRP/Clay Ceramics

Figure 7 shows the SO_2_ reduction rates of all samples. Contrary to the results of NO_2_ absorption tests, in which the concentration reduction rates of Bora and clay ceramic were equivalent, the SO_2_ reduction rate of the clay ceramic was considerably lower than that of the Bora.

The SO_2_ reduction rates of the oxidatively and reductively fired GFRP/clay ceramics were almost equivalent regardless of the difference in the firing conditions and were almost comparable to that of the Bora. Therefore, they were higher than that of the clay ceramic. However, the SO_2_ reduction rates of the ceramics tend to slightly increase with the GFRP mixing ratio.

The above results demonstrate that the GFRP/clay ceramics had a high adsorption capacity for SO_2_ regardless of the firing atmosphere. Based on these results, it can be concluded that the difference in the specific surface area between the oxidatively and reductively fired GFRP/clay ceramics did not significantly affect the SO_2_ adsorption performance of the ceramics. This also indicates that the plastic carbides in the ceramic structure hardly affected the SO_2_ adsorption performance. Therefore, another factor must have had a great influence on the SO_2_ adsorption performance of GFRP/clay ceramics.

## 4. Discussion

### 4.1. The Primary Factor Affecting the NO_2_ Absorption of the Reductively Fired GFRP/Clay Ceramics

To examine the effect of glass fibers and plastic carbide residues on the NO_2_ adsorption performance of the reductively fired GFRP/clay ceramics, NO_2_ adsorption tests were conducted on the glass fibers in the GFRPs and the GFRP carbides. Similar to the previous tests, the particle size of the samples used for NO_2_ adsorption tests was 1.4–2.0 mm. The mass of each sample was also 5 g.

Figure 8 shows the NO_2_ reduction rates of the glass fibers and GFRP carbides. The GFRP carbides exhibited a significantly high NO_2_ adsorption ability. In contrast, that of the glass fibers was considerably low. Thus, the cause of the high NO_2_ adsorption ability of the reductively fired GFRP/clay ceramics is the high NO_2_ adsorption of the plastic carbide residue in the clay structure. The high NO_2_ adsorption of the plastic carbides can be attributed to its high specific surface area [25].

Although we performed the elemental analysis by EDX on the GFRP carbides after the NO_2_ adsorption test, nitrogen was not detected. NO_2_ may be stored in the pores rather than adsorbed on the ceramic surface [21,22,24].

Regarding the NO_2_ adsorption of activated carbon, Urano et al. [25] reported that NO_2_ was adsorbed, without changing into other substances, to the activated carbon under a nitrogen atmosphere; the adsorption mechanism is physical adsorption with a weak binding force. However, NO_2_ was adsorbed as nitric acid (HNO_3_) to the activated carbon under a nitrogen atmosphere containing water vapor.

Regarding the NO_2_ adsorption on the reductively fired GFRP/clay ceramics, since NO_2_ is a polar molecule, it seems that NO_2_ and ceramic surface interacted electrically. Thus, it is presumed that the NO_2_ adsorption mechanism of the ceramics is also physical adsorption.

### 4.2. The Primary Factor Affecting the SO_2_ Absorption of GFRP/Clay Ceramics

Figure 9 shows the SO_2_ reduction rates on the glass fibers and GFRP carbides. The SO_2_ reduction rates of the glass fibers and GFRP carbides were equivalent and slightly higher than that of the Bora.

The result indicates that the primary factor affecting the SO_2_ adsorption of the GFRP/clay ceramics is the content of the glass fibers or plastic carbides. However, the plastic carbides in GFRP carbides represent only 7% of the total mass. Furthermore, the SO_2_ reduction rates of GFRP/clay ceramics (Figure 7) did not depend on the carbon content. This indicates that the primary factor affecting the SO_2_ absorption of the GFRP/clay ceramics is the glass fiber content. This is consistent with the fact that the SO_2_ reduction rates of the GFRP/clay ceramics tend to increase with the increase in the glass fiber content of the ceramics.

Since SO_2_ is also a polar molecule, it may have interacted electrically with the ceramic surface. Thus, it is presumed that the SO_2_ adsorption mechanism of the GFRP/clay ceramics involves a chemical reaction with glass fibers in addition to the interaction with the ceramic surface. Plastic carbides did not contribute to the adsorption.

Next, to verify that SO_2_ was adsorbed onto the surface of the glass fiber, we performed elemental analyses of glass fibers before and after SO_2_ adsorption tests using a SEM with an energy dispersive X-ray spectroscopy (EDS) analysis function (S-5500, Hitachi Ltd., Tokyo, Japan). Figure 10a,b shows the elemental analytical results of glass fibers before and after SO_2_ adsorption tests. Sulfur was detected only in the sample after the SO_2_ adsorption test. Figure 10c and d shows an image of the sulfur elemental mapping via EDS and the SEM, respectively. The sulfur element was locally distributed near the glass fibers.

SO_2_ easily changes to SO_3_ by combining with oxygen in the air. Further, SO_3_ changes H_2_SO_4_ by reacting with moist air. H_2_SO_4_ may react with metals such as calcium to generate sulfates. Therefore, we measured the X-ray diffraction profile of the glass fibers after the adsorption test using an X-ray crystal structure analyzer (PANalytical, X’Pert-Pro MRD, Malvern Panalytical Ltd., Worcestershire, UK); however, sulfate was not detected. The adsorption mechanism of SO_2_ and glass fibers is an issue to be addressed in the future.

## 5. Conclusions

The possibility of using GFRP/clay ceramics as NO_x_ and SO_x_ adsorbents was investigated for an effective reuse of waste GFRPs. The reductively fired GFRP/clay ceramics exhibited a high adsorption performance for both NO_2_ and SO_2_. The primary factor affecting the NO_2_ adsorption of the ceramics was the plastic carbide content in the clay structure, while the glass fiber content was the primary factor affecting the SO_2_ adsorption of the ceramics. Thus, the reductively fired GFRP/clay ceramics could be used as adsorbent materials to remove NO_x_ and SO_x_ from the combustion gases of fossil fuels.

## 6. Patents

Kinoshita H, Kaizu K., Ikeda K., (2013) Manufacturing method of porous ceramics using waste GFRP, Japanese Patent No. 5167520 (in Japanese).

## Figures and Tables

**Figure 1 polymers-14-00164-f001:**
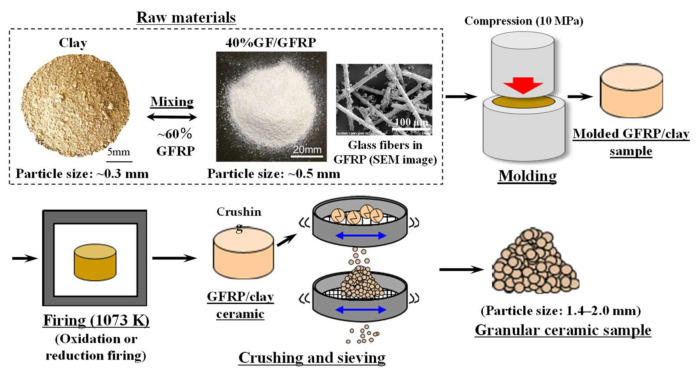
The manufacturing process of GFRP/clay ceramic samples.

**Figure 2 polymers-14-00164-f002:**
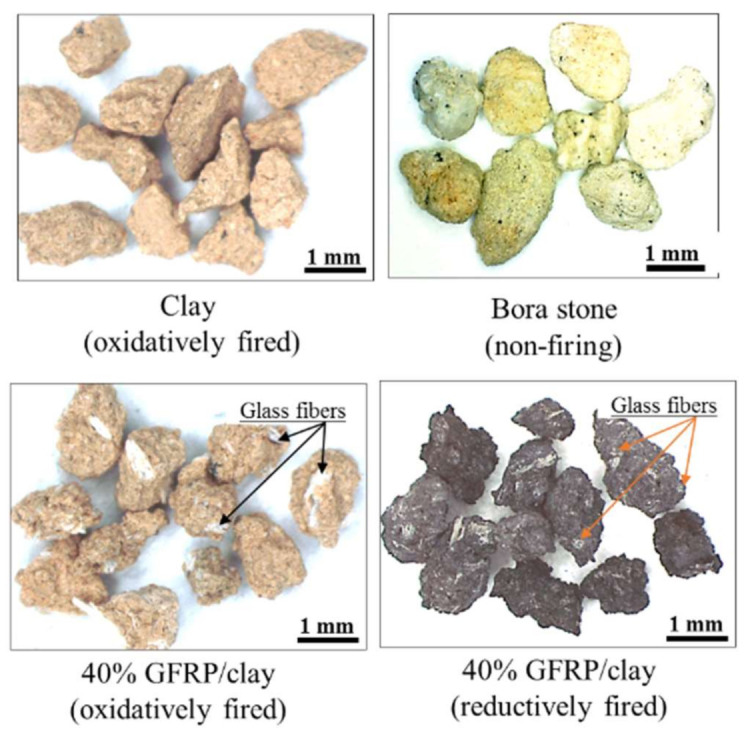
Microscope images of the granular GFRP/clay ceramic specimens.

**Figure 3 polymers-14-00164-f003:**
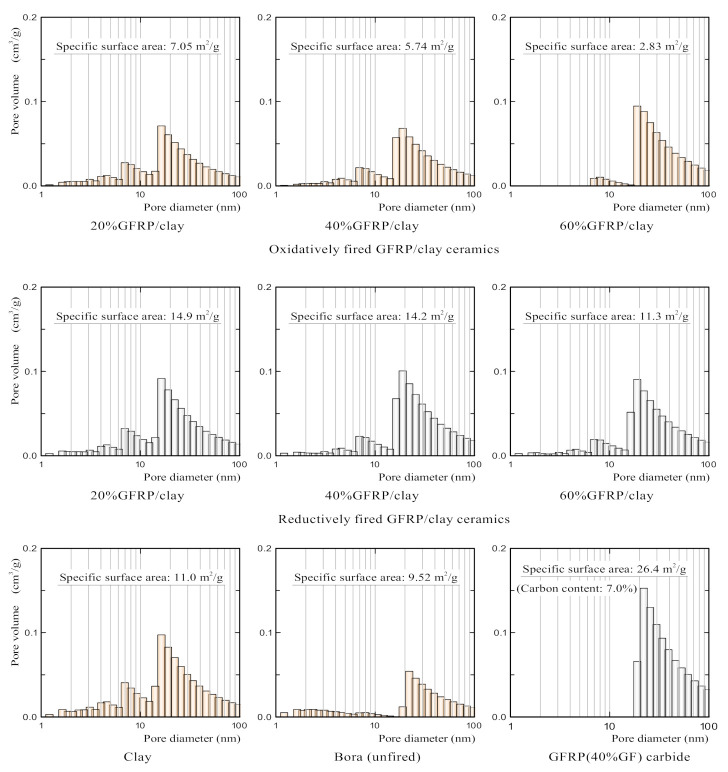
Pore size distributions in the samples.

**Figure 4 polymers-14-00164-f004:**
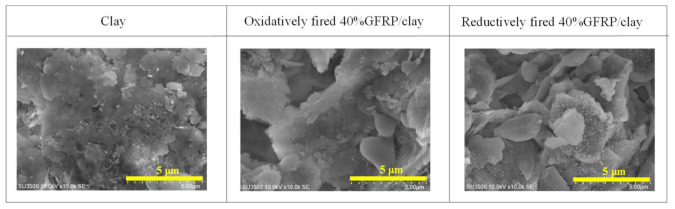
SEM images of the surface structures of clay and GFRP/clay ceramic samples.

**Figure 5 polymers-14-00164-f005:**
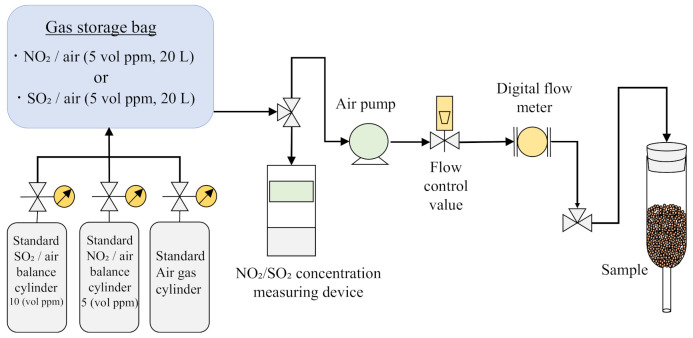
A schematic diagram of the NO_2_ or SO_2_ adsorption test.

**Figure 6 polymers-14-00164-f006:**
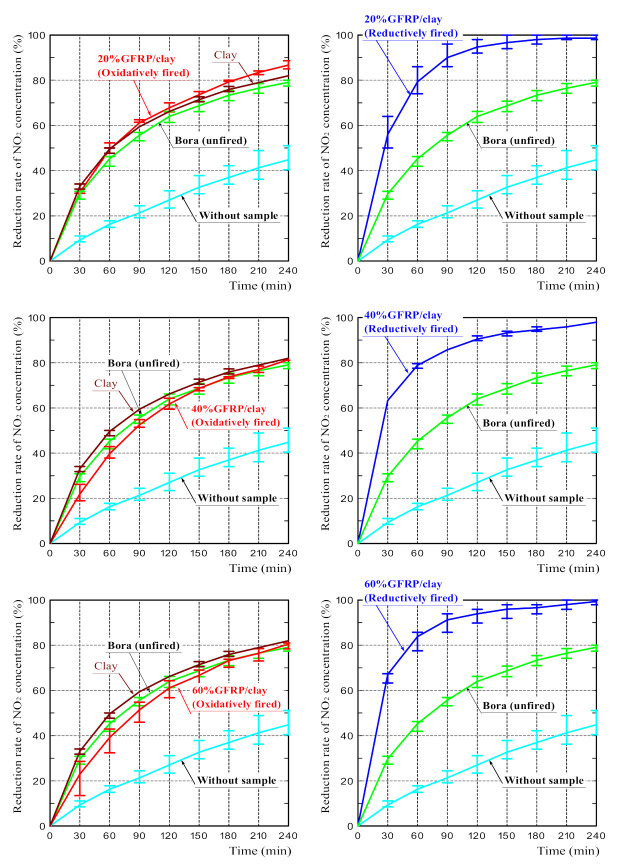
NO_2_ reduction rates for the samples.

**Figure 7 polymers-14-00164-f007:**
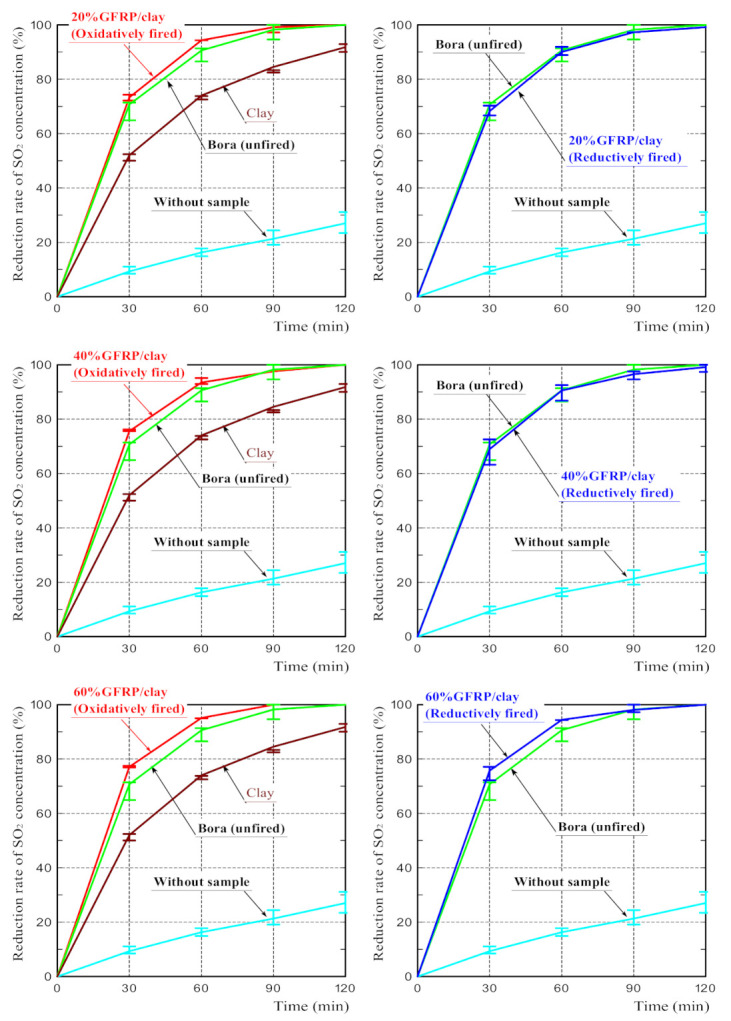
SO_2_ concentration reduction rates for the samples.

**Figure 8 polymers-14-00164-f008:**
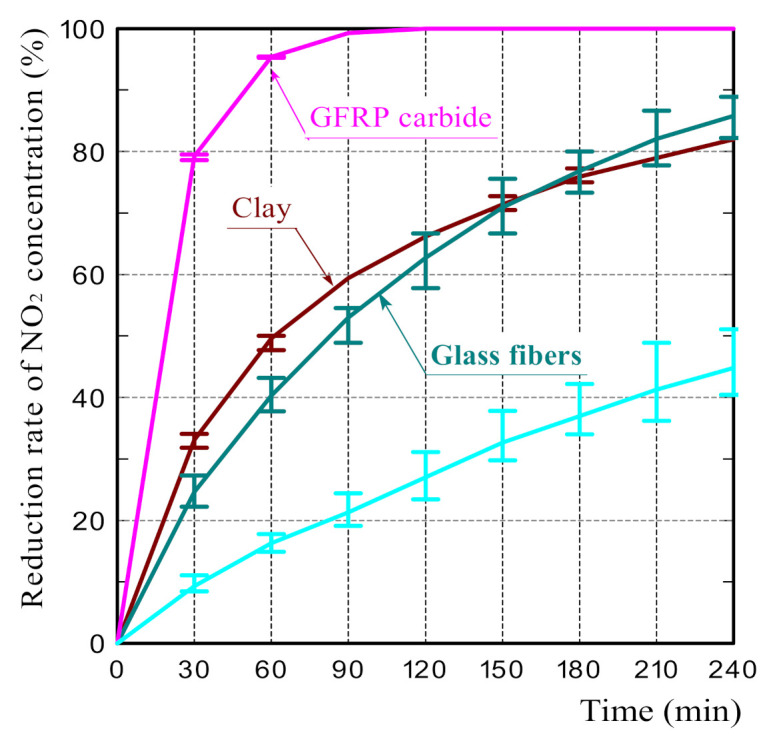
The reduction rates of NO_2_ concentration on the glass fiber and GFRP carbides.

**Figure 9 polymers-14-00164-f009:**
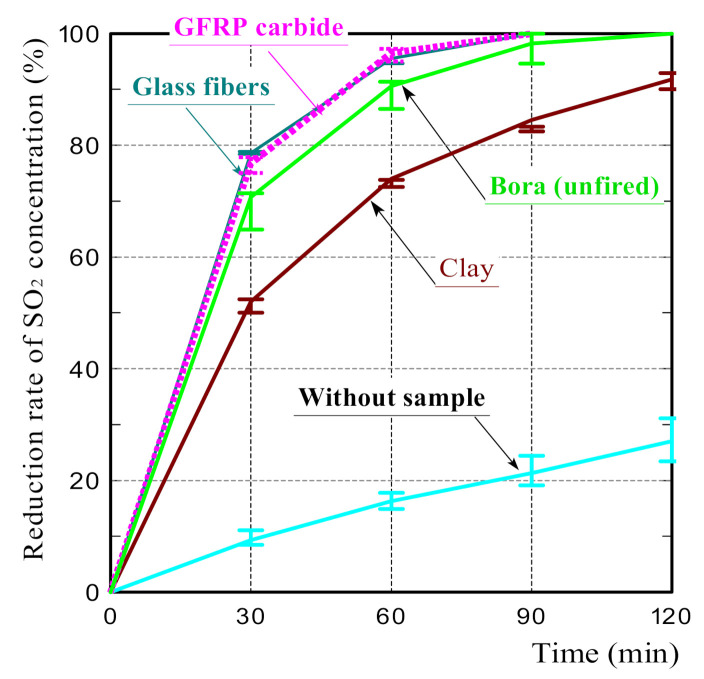
The reduction rates of SO_2_ concentration on the glass fiber and the carbide of GFRP.

**Figure 10 polymers-14-00164-f010:**
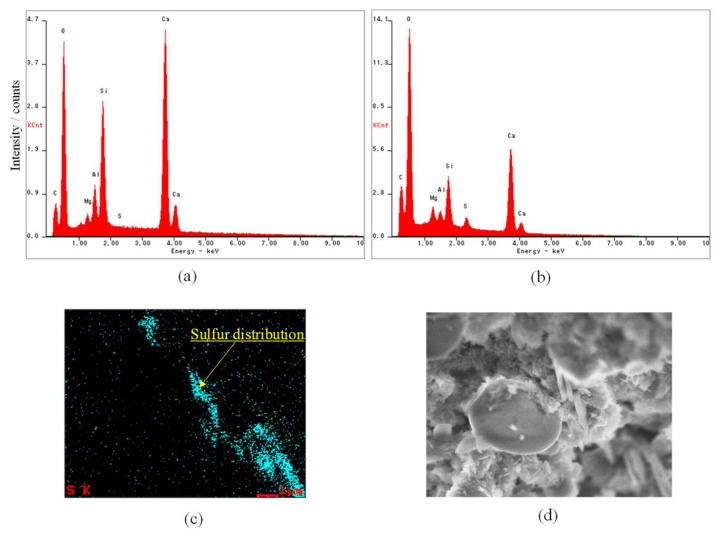
Elemental analysis results of glass fibers before and after SO_2_ adsorption tests. (**a**) Spectral profile before SO_2_ adsorption test, (**b**) Spectral profile after SO_2_ adsorption test, (**c**) An image of sulfur elemental mapping by EDS, (**d**) SEM image of mapping region.

**Table 1 polymers-14-00164-t001:** The compositions of inorganic substances in the glass fiber-reinforced plastic (GFRP), clay, and Bora.

Samples	Component (Mass%)							
	SiO_2_	Al_2_O_3_	Fe_2_O_3_	K_2_O	MgO	CaO	TiO_2_	Others
GFRP (40%GF)	54.9	16.3	0.77	0.15	-	26.7	0.56	0.62
Clay	65.8	21.9	4.79	3.37	1.67	1.31	0.87	0.29
Bora (unfired)	67.2	20.1	5.0	2.98	0.77	3.19	0.55	0.18

**Table 2 polymers-14-00164-t002:** Inorganic chemical compositions of the GFRP/clay ceramic samples.

Samples		Component (Mass%)							
		SiO_2_	Al_2_O_3_	Fe_2_O_3_	K_2_O	MgO	CaO	TiO_2_	Others
Oxidatively fired ceramics	20% GFRP/clay	62.6	22.1	4.87	3.26	1.66	4.02	0.86	0.58
	40% GFRP/clay	59.1	20.7	4.16	2.91	1.75	9.93	0.8	0.71
	60% GFRP/clay	50	17.7	4.09	2.0	1.51	23.2	1.03	0.45
Reductively fired ceramics	20% GFRP/clay	62.2	18.5	6.13	3.73	2.24	5.34	1.21	0.65
	40% GFRP/clay	61.2	9.13	7.56	3.77	2.43	12.9	1.56	1.46
	60% GFRP/clay	56.2	4.79	7.34	3.11	2.14	22.7	1.49	2.22

**Table 3 polymers-14-00164-t003:** Apparent porosity, specific surface area and carbon content of the samples.

Samples		Apparent Porosity (%)	Specific Surface Area (m^2^/g)	Carbon Content (%)
Oxidatively fired ceramics	Clay	31.9	11.0	0.06
	20% GFRP/clay	38.2	7.05	0.24
	40% GFRP/clay	52.7	5.74	0.25
	60% GFRP/clay	62.9	2.83	0.26
Reductively fired ceramics	20% GFRP/clay	43.1	14.9	0.85
	40% GFRP/clay	53.8	14.2	0.99
	60% GFRP/clay	66.2	11.3	1.12
Bora		-	9.52	-
GFRP carbide			26.4	7.0

## Data Availability

The data presented in this study are available on request from the corresponding author.

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
