# Peer review of "Porous Ceramics Adsorbents Based on Glass Fiber-Reinforced Plastics for NOx and SOx Removal"

_polymers, 2021, doi:10.3390/polym14010164_

Round 1

Reviewer 1 Report

In this manuscript, Kinoshita et al studied the NOx and SOx absorbing properties of GFRP/Clay mixture fired at different conditions. The authors also experimentally studied the underlying mechanisms. The results are supported by experimental evidence and can provide useful information for developing waste-derived materials. While this is a nice piece of work, some points are not very clear and should be addressed.

  1. The authors concluded that NO2 is stored in the pores of the structure because of the material’s high specific surface area, not by chemical reaction. NO2 tends to be trapped in the pores but no reactions take place. This is interesting. Can the author provide more details of the mechanism? Could comparing the volume of NO2 absorbed by ceramics to the volume of pores inside the ceramics give some insights?
  2. How are the pore size distributions measured? The measuring procedure is not given. It would also be great if the authors can show some SEM images of the pore structures.
  3. “Thus, the SO2 adsorption mechanism of the GFRP/clay ceramics involves a chemical reaction with glass fibers in addition to physical adsorption through van der Waals forces.”

Can the authors provide the corresponding chemical reaction formula? Moreover, regarding the physical adsorption process due to van der Waals forces, I suggest the authors provide details about this mechanism as well.

  1. There are some minor issues about the writing that need to be revised, for example,

- on page 1, line 26, “because of their high porosity”, what does “their” refer to is not clear.

- on page 2, line 52, “Because of the porous nature of the GFRP/clay ceramics, the authors focused on developing them as raw materials for environmentally friendly products.” The logic of this sentence should be checked.

Author Response

Dear reviewer,

Subject: Submission of revised paper “polymers-1521363”

We wish to express our strong appreciation to you for careful reading our manuscript and for giving useful comments. We have carefully reviewed the comments and have revised the manuscript accordingly. The following is a point-by-point response to the comments.

Comments

  1. The authors concluded that NO2 is stored in the pores of the structure because of the material’s high specific surface area, not by chemical reaction. NO2 tends to be trapped in the pores but no reactions take place. This is interesting. Can the author provide more details of the mechanism? Could comparing the volume of NO2 absorbed by ceramics to the volume of pores inside the ceramics give some insights? How are the pore size distributions measured? The measuring procedure is not given. It would also be great if the authors can show some SEM images of the pore structures.
  2. “Thus, the SO2 adsorption mechanism of the GFRP/clay ceramics involves a chemical reaction with glass fibers in addition to physical adsorption through van der Waals forces.” Can the authors provide the corresponding chemical reaction formula?

Moreover, regarding the physical adsorption process due to van der Waals forces, I suggest the authors provide details about this mechanism as well.

  1. There are some minor issues about the writing that need to be revised, for example, - on page 1, line 26, “because of their high porosity”, what does “their” refer to is not clear.

 Answer

  1. First, we added a method for measuring the pore distribution of the sample.

Page 6, line 184;

“The pore size distributions were measured using the same high-precision gas/vapor adsorption measurement instrument, which was used to measure the specific surface area.”

We also showed SEM images of the surface structures of clay and 40%GFRP/clay ceramics in Figure 4, and added the following sentence.

Page 7, line 200;

“Figure 4 shows examples of scanning electron microscope (SEM, S5500, Hitachi High-Technologies Corporation, Tokyo, Japan) images of the surface structures of the clay and 40% GFRP/clay ceramic samples. Further, the GFRP/clay ceramics had more pores than clay ceramic.”

Furthermore, regarding NO2 adsorption mechanism, we have revised the following sentence and added reference [44].

Page 11, line 301;

“Although we performed the elemental analysis by EDX on the GFRP carbides after the NO2 adsorption test, nitrogen was not detected. NO2 may be stored in the pores rather than adsorbed on the ceramic surface [21,22,24].

Regarding the NO2 adsorption of activated carbon, Urano et al [44] reported that NO2 was adsorbed, without changing into other substances, to the activated carbon under a nitrogen atmosphere; the adsorption mechanism is physical adsorption with a weak binding force. However, NO2 was adsorbed as nitric acid (HNO3) to the activated carbon under a nitrogen atmosphere containing water vapor.

Regarding the NO2 adsorption on the reductively fired GFRP/clay ceramics, since NO2 is a polar molecule, it seems that NO2 and ceramic surface interacted electrically. Thus, it is presumed that the NO2 adsorption mechanism of the ceramics is also physical adsorption.”

2. I guess the chemical reaction between glass fibers and SO2 as follows so far:

Sulfur dioxide easily changes to sulfur trioxide by combining with oxygen in the air.

2SO2 + O2 → 2SO3

The sulfur trioxide changes sulfuric acid by reacting with moist air.

SO3 + H2O → H2SO4

Furthermore, calcium sulfate is generated by the reaction of sulfuric acid with calcium.

H2SO4 + Ca (OH) 2 →CaSO4 +2H2O

However, we did not write the chemical reaction formula because we could not demonstrate above reaction by XRD analysis on the glass fibers after SO2 adsorption test. Regarding the adsorption mechanism of SO2 and glass fibers, we would like to treat that as an issue to be addressed in the future.

We added the following sentence.

Page 12, line 342;.

“The adsorption mechanism of SO2 and glass fibers is an issue to be addressed in the future.”

Regarding the physical adsorption of SO2, the interaction between SO2 and the ceramic surface may be mainly due to electrical attraction rather than van der Waals force because SO2 is a polar molecule. So, we revised the sentence as follows.

Page 11, line 326;

“Since SO2 is also a polar molecule, it may have interacted electrically with the ceramic surface. Thus, it is presumed that the SO2 adsorption mechanism of the GFRP/clay ceramics involves a chemical reaction with glass fibers in addition to the interaction with the ceramic surface.”

3. We have corrected some English sentences.

We are grateful for the time and energy you expended on our behalf.

Sincerely yours,

Hiroyuki Kinoshita

University of Miyazaki, Japan

Reviewer 2 Report

This work presented very interesting results on the adsorption of SOx and NOx by using GFRP/clay composites. This research has very high importance, especially considering high issues related to environmental protection. The authors analyzed the influence of GFRP loading, annealing and compared the results to different adsorbents. 

The minor comments are the following:

1) Please check again grammar and spelling and style. Some sentences are hard to read.

2) Please adjust the picture of EDX mapping. Please enhance brightness nd contras.

Author Response

Dear reviewer,

Subject: Submission of revised paper “polymers-1521363”

We wish to express our strong appreciation to you for careful reading our manuscript and for giving useful comments.

Comments

1) Please check again grammar and spelling and style. Some sentences are hard to read.

2) Please adjust the picture of EDX mapping. Please enhance brightness and contrast.

Answer

  • We have corrected some English mistakes.
  • We adjusted the brightness and contrast of the picture of EDX mapping in Fig.10.

We are grateful for the time and energy you expended on our behalf.

Sincerely yours,

Hiroyuki Kinoshita

University of Miyazaki Japan
